# Nanoparticle Delivery Platforms for RNAi Therapeutics Targeting COVID-19 Disease in the Respiratory Tract

**DOI:** 10.3390/ijms23052408

**Published:** 2022-02-22

**Authors:** Yuan Zhang, Juhura G. Almazi, Hui Xin Ong, Matt D. Johansen, Scott Ledger, Daniela Traini, Philip M. Hansbro, Anthony D. Kelleher, Chantelle L. Ahlenstiel

**Affiliations:** 1Kirby Institute, UNSW, Sydney, NSW 2052, Australia; yuanzhang@kirby.unsw.edu.au (Y.Z.); sledger@kirby.unsw.edu.au (S.L.); akelleher@kirby.unsw.edu.au (A.D.K.); 2Respiratory Technology, Woolcock Institute of Medical Research, Sydney, NSW 2037, Australia; juhura.almazi@sydney.edu.au (J.G.A.); huixin.ong@mq.edu.au (H.X.O.); daniela.traini@mq.edu.au (D.T.); 3Macquarie Medical School, Faculty of Medicine, Health and Human Sciences, Macquarie University, Ryde, NSW 2109, Australia; 4Centre for Inflammation, Faculty of Science, Centenary Institute and University of Technology Sydney, Sydney, NSW 2050, Australia; matt.johansen@uts.edu.au (M.D.J.); philip.hansbro@uts.edu.au (P.M.H.)

**Keywords:** COVID-19, nanomedicine, lipid nanoparticles, polymer nanoparticles, glycogen nanoparticles, siRNA, nanoparticle-capsulated drug delivery, inhalation

## Abstract

Since December 2019, a pandemic of COVID-19 disease, caused by the severe acute respiratory syndrome coronavirus 2 (SARS-CoV-2), has rapidly spread across the globe. At present, the Food and Drug Administration (FDA) has issued emergency approval for the use of some antiviral drugs. However, these drugs still have limitations in the specific treatment of COVID-19, and as such, new treatment strategies urgently need to be developed. RNA-interference-based gene therapy provides a tractable target for antiviral treatment. Ensuring cell-specific targeted delivery is important to the success of gene therapy. The use of nanoparticles (NPs) as carriers for the delivery of small interfering RNA (siRNAs) to specific tissues or organs of the human body could play a crucial role in the specific therapy of severe respiratory infections, such as COVID-19. In this review, we describe a variety of novel nanocarriers, such as lipid NPs, star polymer NPs, and glycogen NPs, and summarize the pre-clinical/clinical progress of these nanoparticle platforms in siRNA delivery. We also discuss the application of various NP-capsulated siRNA as therapeutics for SARS-CoV-2 infection, the challenges with targeting these therapeutics to local delivery in the lung, and various inhalation devices used for therapeutic administration. We also discuss currently available animal models that are used for preclinical assessment of RNA-interference-based gene therapy. Advances in this field have the potential for antiviral treatments of COVID-19 disease and could be adapted to treat a range of respiratory diseases.

## 1. Introduction

According to data by WHO (https://covid19.who.int/), as of 31 January 2022, over 373 million people worldwide have been infected with SARS-CoV-2, and there have been more than 5.6 million confirmed deaths due to this virus since December 2019. At present, the response strategy to COVID-19 mainly relies on vaccination. Current COVID-19 vaccines are designed to protect the body from infection through a strong antibody response [1,2]. However, a recent UK study examined the serum of COVID-19 patients for 94 days following SARS-CoV-2 infection and reported that antibody levels peaked at 3–4 weeks post infection and then decreased exponentially. Meanwhile, peak antibody titers positively correlated with the time it took for antibodies to return to baseline levels [3]. Another study also demonstrated that antibody responses were short-lived following COVID-19 infection [4]. Further, the immune response to available COVID-19 vaccines drop at 3 and 6 months after the second dose of vaccination, and antibody levels to the vaccines negatively correlated with the age of vaccinated individuals [5]. Moreover, vaccination is likely to be suboptimal in those with impaired immunity and allergies and be further compromised by the emergence of variants of concern (VOC). Vaccination may not provide adequate protection for people with a dysfunctional immune system. Therefore, to effectively reduce the risk of serious illness and hospitalization, it is necessary to develop antiviral treatments against COVID-19.

All direct-acting antiviral therapies that are active against SARS-CoV-2 need to be given early in the disease course, so they need to be administered easily and should be safe [6]. Several drugs are being tested for the treatment of COVID-19, which include antiviral, anti-malarial, and anticancer agents. Notably, the latest two orally delivered antiviral drugs, the Pfizer Paxlovid (a protease inhibitor) and Merck molnupiravir (a polymerase inhibitor) have been proven to be effective in treating SARS-CoV-2 infection [7,8,9]. Particularly, Paxlovid might have therapeutic potential in the newly emerging Omicron variant [10], which is fundamentally different from previous VOC, as it does not appear to use the TMPRSS2 protease co-receptor [11,12] or target the lung as effectively as previous ones [13,14,15,16,17]. As there is currently no specific treatment for SARS-CoV-2, drugs for the treatment of other viral infections, such as Remdesivir, Baricitinib, Tocilizumab, and Favipiravir, have been repurposed and may be used to assist in the treatment of COVID-19 [18]. Some of these antiviral drugs have already been shown to have limitations in treating SARS-CoV-2, and the side effects of those antiviral drugs on COVID-19 patients also remain to be explored [19]. Other available therapeutics such as the intravenous (IV) injection of monoclonal antibodies (mAb) are highly efficacious; however, production is intensive, time-consuming, and expensive [20]. Moreover, the therapeutic efficacy of mAbs is significantly affected by a virus with a high mutation rate, and it is difficult to perform organ-targeted therapy [21]. Thus, the therapeutic effects and side effects of these currently available antiviral drugs on COVID-19 patients remain to be defined [19]. Since many current treatments for COVID-19 patients are repurposed therapeutics for other diseases, investigating approaches that are targeted and specific to COVID-19 disease is highly valuable. One potential treatment strategy is RNA interference (RNAi), and a more targeted drug delivery platform involves nanoparticle (NP) carriers.

RNAi is a conserved biological process induced by non-coding RNAs (ncRNAs) with the mechanism of inhibiting gene expression by blocking the transcription or translation of specific genes. This phenomenon was first discovered and proposed in Caenorhabditis elegans by Fire et al. [22]. Small RNAs that can produce RNAi include small interfering RNAs (siRNAs), microRNAs (miRNAs), P-element-induced wimpy testis (PIWI)-interacting RNAs (piRNAs) and endogenous siRNAs (esiRNAs) [23]. As a phenomenon that has been widely studied, RNAi has been proven to be a potential treatment strategy for many diseases (viral and bacterial infections, respiratory diseases, cancer, and autoimmune diseases), including COVID-19 [24,25,26]. Currently, the most studied non-coding RNAs are siRNA (exogeneous and double-stranded) and miRNA (endogenous and single-stranded), which play a vital role in gene regulation [27]. The physical and chemical properties of siRNA and miRNA are similar, while the mechanisms underlying their mode of action are different, and their stability and targeted delivery in vivo also have challenges [28]. Studies have found that both siRNA and miRNA have the potential to be used in treating COVID-19 [29]. Treatments directly into the lung are relatively stable for miRNAs [27]. Importantly, siRNA can be also delivered to the lung epithelium of mice via intranasal or intravenous administration via liposomes [30,31]. This review focuses on summarizing the therapeutic role of siRNA in COVID-19 disease. Furthermore, siRNA could inhibit viral infection by silencing specific genes that are required for viral entry or utilized for viral replication [32,33]. A variety of siRNAs that target highly conserved regions of SARS-CoV-2 sequence have been identified [30,34]. Delivery systems for these siRNAs can be divided into viral vectors and nanoparticles [35,36]. Viral delivery systems, such as adenovirus and lentivirus, are the most used viruses in research and can successfully deliver siRNA in vivo [37,38,39]. However, several disadvantages of these vectors are known, such as their high immunogenicity [40], restriction of specific delivery to target cells [41], the need for ex vivo transduction and reinfusion [42], limited transduction rates for certain cell types [43], potential mutagenicity of integration sites [44], and complexity and high costs of GMP synthesis [45]. By contrast, several bioconjugates have been applied to carry siRNAs in vitro and in vivo, including immune proteins, peptides, aptamer, cholesterol, PEG, cell-penetrating peptides, and NPs.

NPs have been identified as carriers to deliver functional nucleic acid fragments into the body for specific targeted therapy [36], which has recently become a new medical field [46,47,48,49,50]. Compared with viral delivery systems, NPs can be designed to have low or limited immunogenicity. Another advantage of NP carriers is that they can be designed to ensure that siRNA is protected from ribonuclease (RNase), making it stable and difficult to degrade by enzymatic degradation. NP carrier systems can be in the form of vesicle-based NP suspensions after encapsulation [51]. Thus, NPs can be used as a carrier platform to encapsulate therapeutic drugs and have played an increasingly promising role in the targeted therapy of diseases [52,53,54].

So far, siRNA treatment measures have mostly targeted the liver, as this organ represents an accessible target due to the first-pass metabolism of both siRNA and many nanoparticles. ONPATTRO™ (patisiran), approved by the US Food and Drug Administration (FDA) and the European Medicines Agency (EMA), also uses lipid NPs to effectively deliver siRNAs to hepatocytes for therapeutic effect [55,56,57]. Since SARS-CoV-2 has been found mainly in the respiratory tract of COVID-19 patients [58], a logical treatment strategy is to deliver therapeutics directly to the lung and respiratory tract via topical administration or administration directly into the airways, avoiding the high first-pass metabolism of systemically delivered siRNAs by the liver [59]. Therefore, combining the targeting properties of NPs with the specificity of RNAi could allow for the design of products with better therapeutic efficacy with lower side effect profiles. Herein, we discuss a variety of potential nanocarriers, including organic nanoparticles, such as lipid NPs, polymer NPs, and glycogen NPs, and inorganic nanoparticles, such as gold NPs and magnetic NPs; then, we summarize the progress of these platforms in siRNA delivery. We also present the potential of formulating NP-siRNAs that specifically target the lungs or respiratory tract and evaluate the potential application of various NP-capsulated siRNAs (NP-siRNAs) as therapeutics for SARS-CoV-2 infection.

## 2. siRNA Therapy—A Potential and Promising Antiviral Therapeutic

It is well known that siRNAs are derived from long double-stranded RNAs, which are cleaved into small double-stranded RNA with the same length of ~21 nucleotides through endonuclease Dicer-mediated cleavage. siRNAs have the potential to act as a highly specific therapeutic strategy for the treatment of various diseases [46,60,61]. So far, the FDA has approved four siRNA therapies, including patisiran (for polyneuropathy of hereditary TTR-mediated amyloidosis) [57], givosiran (for acute hepatic porphyria) [62], lumasiran (for primary hyperoxaluria type 1) [63], and inclisiran (for primary hypercholesterolemia or mixed dyslipidemia) [64]. siRNA therapy has made great progress in tumor treatments [65]. siRNA therapeutics for some cancers were developed from animal experiments through to clinical trials. A study in mice demonstrated that siRNAs were effective in reducing melanoma growth [66]. Additionally, some siRNAs (such as CALAA-01, Atu027, and DCR-MYC) have been used in phase I clinical trials in metastatic pancreatic cancer, colon cancer, and hepatocellular carcinoma [67,68,69,70,71], and siG12D-LODER has been used in phase II clinical trials in advanced pancreatic cancer [72]. Some studies have also revealed that siRNAs can be used to combat viral infections, especially COVID-19 [73,74].

### 2.1. Mechanisms of siRNA Therapy

siRNAs can induce gene silencing in mammalian cells through two distinct pathways; post-transcriptional gene silencing (PTGS) or transcriptional gene silencing (TGS) (Figure 1). PTGS primarily occur in the cytoplasm. Firstly, the guide strand of the siRNAs binds to the Argonaute-2 protein (Ago-2) of the RNA-inducing silencing complex (RISC), while the other non-guide strand is cleaved and ejected by the Ago-2 [75]. Subsequently, Ago-2 carries the guide strand to target mRNA complementary to its sequence, causing the degradation of the binding site of the target RNA to induce gene expression silencing. Finally, RISC ejects the guide strand in preparation for the next round of gene silencing [76]. PTGS is considered the main pathway of siRNA-mediated mRNA silencing [77]. TGS is another pathway of siRNA-induced gene silencing. It is less well characterized in mammalian cells and takes place exclusively in the nucleus [78]. Unlike PTGS, in TGS, siRNA binds to the Ago-1 protein on the RNA-induced transcriptional silencing (RITS) complex [79]. Increasing the methylation and decreasing the acetylation of critical chromatin-associated proteins, contributes to the production of heterochromatin and induces the epigenetic silencing of the gene [80]. TGS is potentially more beneficial for the long-term treatment of chronic infections, whereas the PTGS is likely adequate for acute infections such as COVID-19.

### 2.2. siRNA Therapeutics for SARS-CoV-2, SARS-CoV-1, and MERS-CoV

Among all human coronaviruses (CoV), there are now seven main types that cause respiratory infections. Four of these are the previously prevalent CoVs (229E, OC43, NL63, and HKU1) which cause mild respiratory infections, while the remaining three CoVs (MERS -CoV, SARS-CoV-1, and SARS-CoV-2) can cause severe viral pneumonia, including severe acute respiratory syndrome (SARS), Middle East respiratory syndrome (MERS), and COVID-19 [81,82].

CoVs are a group of roughly spherical virus particles with a mean diameter of 80–120 nm. Their viral envelope consists of the membrane (M), envelope (E), and spike (S) proteins, and the nucleocapsid inside the envelope is mainly composed of nucleocapsid (N) protein [83]. The CoV genome contains a positive-sense single-stranded RNA, which has a genomic structure of 5′UTR-ORF1ab-S-E-M-N-3′UTR-poly (A) tail. Open reading frames (ORFs), which are not only necessary during virus replication [84], but also play a role in the pathogenicity and infectivity of the virus [85], occupy nearly two-thirds of the entire viral genome and encode the replicase polyprotein (pp1ab). The subsequent reading frames encode the major structural proteins (S, E, M, and N proteins) [86]. The severity of CoV disease varies widely. MERS-CoV, SARS-CoV, and SARS-CoV-2 are three major pathogenic viruses that can cause severe symptoms [87]. SARS-CoV-1 and SARS-CoV-2 share multiple highly homologous regions. Several B and T cell epitopes are highly conserved between them, which indicates that vaccines and therapeutic strategies designed to target these conserved regions could contribute to the prevention of SARS-CoV-1/2 infection to some extent [88]. In terms of the genetic sequence, the similarity of SARS-CoV-2 to SARS-CoV-1 and MERS-CoV is approximately 79% and 50%, respectively [89]. When codon bias occurs, one codon can be more efficient than others during translation. Compared to SARS-CoV-1 and MERS-CoV, SARS-CoV-2 is capable of enhanced gene expression due to its higher codon usage bias [90].

Several factors need to be considered to improve the effectiveness of siRNAs. Firstly, their binding targets should be highly conserved to decrease the possibility of viral escape caused by viral evolution [91]. For the treatment of SARS-CoV-2 infection, it is suggested that siRNA therapeutics that target the highly conserved region of SARS-CoV-2 could be effective [35]. siRNAs designed against SARS-CoV-2 infection mainly act on coding regions for cellular invasion factors (e.g., host factors ACE2, TMPRSS2, and spike) and gene sequences required for viral survival and amplification (e.g., RNA-dependent RNA polymerases (RdRP)) [24]. Meanwhile, the cell protease furin has been found to activate the S protein of SARS-CoV-2 through cooperation with TMPRSS2 during virus invasion, which makes furin another target of siRNAs [92]. The 5′-UTR of the SARS-CoV-2 genome contains a leader sequence, which could also become an effective target for the development of siRNA-specific viral therapies [34].

Apart from identifying the target sequence of SARS-CoV-2, it is also necessary to avoid off-target effects as well as optimize the gene-silencing function of siRNA [93]. During the design of siRNAs, homology and specificity checks should be performed to avoid off-target effects. At the same time, it is also necessary to avoid acidification and degradation of siRNA after it enters the cell to form an endocytic vesicle [73]. In addition, siRNA formulation (i.e., GC content) also affects the silencing effect. Lower GC % will result in decreased specificity and weak binding, while higher GC % will hinder the unwinding of its duplex [94].

Studies show that some siRNAs are effective against SARS-CoV-1 [95,96,97,98] and MERS-CoV in vitro [84,99,100]. A computational model was applied to evaluate the application and potential effectiveness of RNAi in MERS-CoV and found that five siRNAs and four miRNAs are potential antiviral agents [99]. Another study designed and evaluated the functions of six siRNAs with increased specificity. These siRNAs could combine SARS-CoV-2 mRNA targets, inducing greater inhibition of viral replication and reduced off-target effects (Table 1) [91]. In addition, bioinformatics-based methods were also used to design siRNA targeting SARS-CoV-2 RdRp as a therapeutic agent [101]. Chen et al. identified nine potential siRNA targets in the SARS-CoV-2 genome [102]. They found that an effective siRNA (5′GUUUAGAGAACAGAUCUACAA3′) has a high affinity to the leader sequence and a low possibility of off-target effects [103]. Moreover, 11 siRNAs were designed and tested for silencing efficiency against the conserved regions of the three viral genes, the spike (S), nucleocapsid (N), and membrane (M). This revealed that four siRNAs (3329i, 1878i, 1104i, and 2351i) were able to decrease the expression of the S gene, and four other siRNAs (418i, 881i, 214i, and 1068i) could separately reduce the expression of the N gene, while the remaining three siRNAs (607i, 344i, and 79i) could decrease the M gene expression [104]. Several researchers have also predicted functional and potential siRNAs as therapeutics using in silico analysis [105,106] (Table 1).

Although RNAi therapeutics may have a positive influence in fighting COVID-19, there remain challenges in the safe delivery of siRNA [107]. Therefore, various improvements concerning delivery platforms need to be developed to manage this problem.

## 3. NP Delivery—A Platform Ensuring the Efficacy and Stability of siRNAs

NPs are valuable carriers to deliver functional nucleic acid fragments into the body for specific targeted therapy. These nanocarrier systems can be used to form vesicle-based NP suspensions after encapsulation, thereby improving the stability of the small-molecule drugs contained therein [51]. Thus far, the four siRNA drugs approved by the FDA applied N-acetylgalactosamine (GalNAc) coupling or lipid NP pathways to effectively deliver siRNAs to liver cells for their therapeutic effects. NPs can also improve the therapeutic effects of drugs by changing drug metabolic rates [121]. NPs can be divided into different types according to their size, shape, and physical/chemical properties. Advanced nanocarriers include organic NPs (lipid NPs, polymer NPs, and glycogen NPs) and inorganic NPs (magnetic NPs, gold NPs, carbon-based NPs, ceramic NPs, metal NPs, and semiconductor NPs) (Figure 2) [122,123].

### 3.1. Lipid NPs (LNPs) in siRNA Therapeutics

LNPs belong to the organic nano-system and have hydrophilic parts exposed to the surrounding aqueous solvent as well as hydrophobic parts facing each other to form a supramolecular structure [124]. LNPs are particularly attractive for biomedical applications, owing to their enhanced lipid-specific biocompatibility. Thus, LNPs are widely used in the field of nanomedicine. In 2018, the FDA and the EMA approved a therapeutic drug for hereditary transthyretin-mediated amyloidosis (hATTR amyloidosis) based on siRNA LNPs, ONPATTRO™ (patisiran) [55,56]. In patisiran, the weight ratio of total lipids and siRNAs is 12.1, and the composition of these LNPs, which could effectively target the transthyretin of hepatocytes, includes ionizable lipid, phospholipid, cholesterol, and PEG lipid [125]. To date, LNPs are the most frequently studied in vivo delivery carriers, and they can silence target genes by encapsulating and delivering specific siRNAs. LNPs can also ensure the stability of siRNA in vivo, protecting them from degradation and ensuring smooth delivery to target cells.

Liposomes, as the earliest generation of lipid particles, are small-molecule artificial vesicles that are spherical in structure, with diameters of 25–2500 nm. According to their size, they can be divided into multilamellar (MUV) or unilamellar vesicles (UV). UV can be divided into LUV (large unilamellar vesicles, 100–1000 nm in diameter) and SUV (small unilamellar vesicles, 20–100 nm in diameter). Liposomes consist of an amphiphilic bilayer phospholipid on the outside and an aqueous compartment in the center (Figure 3A) [123]. The bilayer phospholipid can carry hydrophobic therapeutic drugs in the middle, and the aqueous compartment can carry hydrophilic therapeutic drugs internally [126].

Compared with classic liposomes, cationic LNPs have a more complex liposome-like structure and are more suitable for encapsulating various nucleic acids (RNA and DNA). Cationic LNPs, spherical particles ~50 nm in diameter, are composed of ionizable cationic lipids, cholesterol, phospholipids, and polyethylene glycol-lipids (PEGylated lipids) (Figure 3B) [127]. Various lipids in LNPs have different characteristics and functions. Cationic lipids can also ensure the stability of siRNA. Ionizable cationic lipids can be ionized under acidic conditions (pH < 5), encapsulating siRNA in particles through electrostatic complexation. Subsequently, when the positively charged cationic lipid contacts the negatively charged cell membrane, it can be endocytosed into the cell and release the siRNA inside [128,129]. PEGylated lipids are used to control the spacing between NPs, inhibit LNP aggregation, and prevent them from being recognized by the human immune system [130]. Neutral lipids promote and stabilize the formation and arrangement of the phospholipid bilayer structure, while cholesterol has strong membrane fusibility, which can promote the intracellular uptake and cytoplasmic entry of siRNA [131,132].

The ratio of polyethylene glycol (PEG) to LNPs can affect the efficacy of siRNAs in gene silencing [133], while increasing the size of LNPs may help redirect siRNAs in them to antigen-presenting cells (APCs) [134]. The release of siRNA from LNPs is considered a rate-limiting factor in the process of gene silencing. Therefore, it is necessary to increase the release-potential of siRNA to promote gene silencing [135].

siRNA LNPs have the potential to play a vital role in the treatment of many diseases. Intravenous injection of anti-miRNA145 LNPs in a rat model of pulmonary arterial hypertension (PAH) improved heart function [136]. It was also found that siRNA LNPs can enter bone cells after intravenous injection to induce the gene silencing of SOST encoding sclerostin, providing a new possibility for the treatment of osteoporosis [137]. New research has developed a novel type of siRNA LNP delivery system (LNP-Ihh siRNA). Deleting cartilage-specific genes (Ihh) to reduce cartilage degeneration is hoped to provide support for the treatment of cartilage diseases [138].

The delivery of siRNAs by LNPs may also play an important role in the treatment of tumors and blood diseases. A study applied tripeptide LNP as a carrier and co-wrapped chemotherapeutic drug paclitaxel (PTX) and VEGF siRNA with sucrose laurate (S) and folate-PEG2000-DSPE (FA) to construct a new type of nanoparticle (PTX/siRNA/FALS), which could target tumor cells to improve the efficacy of cancer treatment [139]. Moreover, LNPs can encapsulate multiplexed siRNA and play a positive role in the treatment of lymphoma [140]. Packaging a leukemia-specific fusion transcript, BCR-ABL, and siRNA into LNPs can also be used to target oncogenes in hematopoietic tissue in vitro and in vivo [141]. In addition, some helper lipids such as DSPC-cholesterol could facilitate the stable delivery of siRNA by LNPs [142].

Currently, ionizable LNPs are recognized as one of the most advanced non-viral vectors for efficient nucleic acid delivery [143]. Idris et al. identified three candidate siRNAs from numerous siRNAs targeting the conserved regions of SARS-CoV-2, and their inhibitory effect on the virus was as high as 90%. At the same time, two new LNPs that can deliver siRNAs to the lung have also been developed [30], and after intranasal administration, cationic LNPs show higher bioavailability [144]. LNPs also have the potential to load vaccines [131,145,146,147].

In addition, other types of LNPs have also been used to deliver siRNA, including solid LNPs (SLNs) [148], nanostructured lipid carriers (NLCs) [149], nonlamellar LNPs (NLNs) [150], ethosomes [151], and echogenic liposomes [152] (Table 2). These LNPs could also improve the stability of siRNA and effectively release siRNA after entering the target cells [127]. Therefore, LNPs have been recognized as the most effective siRNA delivery system.

### 3.2. Polymer NPs (PNPs) in siRNA Therapeutics

PNPs are small molecular particles with a size of 1–1000 nm and include nanospheres and nanocapsules. PNPs are used for drug or small RNA delivery and are generally biodegradable polymers, in which the poly(lactic acid) (PLA), poly(glycolic acid) (PGA), and poly (lactic-co-glycolide) (PLGA) have been approved by FDA [172]. They are considered to have good safety and biocompatibility, as well as low levels of immunogenicity and toxicity [173].

Earlier, fluorescent conjugated PNPs (cPNPs), which were found to deliver gene-specific siRNA to tobacco BY-2 protoplasts [174], are considered non-toxic and efficient in the delivery of siRNA [175]. siRNA PNPs can be used to deliver various tumor drugs as well as siRNAs/miRNAs to assist in cancer treatment [176]. A study developed cationic core–shell NPs, called P(MDS-co-CES), which may be used to deliver both chemotherapeutic drugs (paclitaxel) and siRNA to the same cell [177]. The PEGylation of siRNA reduces the protein adsorption of the PNP–siRNA complex, thereby delivering siRNA more efficiently [178]. Multifunctional poly (lactide-co-glycolic acid) (PLGA) NPs have also been developed to deliver tumor antigen and immunosuppressive gene (SOCS1) siRNA to bone-marrow-derived (BM) dendritic cells (DC) to enhance the effects of immunotherapy [179]. Doxorubicin (DOX) and siRNA could be co-loaded into PNPs containing polyethylenimine (PEI) and then delivered to the lungs, which could be effective in the treatment of metastatic lung cancers [180].

Meanwhile, encapsulated siRNA can also be assembled using star polymers, which have a multi-branched structure and can self-assemble with siRNA to form cationic nano-sized complexes to enhance cell entry. Star polymer–siRNA can hold the rate of targeted gene silencing at 50%. Kavallaris et al. applied poly (DMAE-MA) loaded with Luc2 siRNA to human pancreatic cancer (MiaPaCa-2) and H460 non-small cell lung cancer (NSCLC) cell lines and found that star polymer–siRNA can inhibit the expression of target genes in both [181]. In BALB/c nude mice transplanted with human pancreatic cancer cells, star–POEGMA–siRNA can be delivered to mouse pancreatic cancer cells [182]. Subsequently, in the BALB/c nude mice model of pancreatic cancer in situ, the use of star polymer loaded with TUBB3 (βIII-tubulin) siRNA for treatment can silence the expression of βIII-tubulin and inhibit the growth and metastasis of pancreatic tumors [183]. In addition, star polymer–siRNA can also be used to enhance the drug delivery potential in acute lymphoblastic leukemia (ALL) cells [184].

Interestingly, NPs that integrate lipids and polymers (hybrid lipid–polymer NPs) were found to combine the biocompatibility of lipids with the structural solidity of polymers and enhance the binding affinity with siRNA to maintain stable encapsulation of siRNA [185]. Hybrid lipid–polymer NPs with siRNA exhibited excellent gene silencing effects in HeLa and A549 lung carcinoma cells [186]. Another study also found that applying lipid-assisted PNPs could efficiently deliver specific siRNA targeting GLUT3 into glioma stem cells and bulk cancer cells, thereby enhancing the effectiveness of cancer treatment [187].

As carriers, PNPs also play an important role in the delivery of therapeutic drugs and siRNA for metabolic diseases (such as diabetes) and infectious diseases (such as reproductive tract and severe lung diseases). Biodegradable PLGA NPs loaded with siRNA could enter the mucosal epithelial tissue of the reproductive tract to play a therapeutic role against infectious pathogens [188]. Cationic PNPs composed of β-cyclodextrin (β-CD) and poly amidoamine can be used as an effective carrier of MMP-9-siRNA and injected into the wounds of diabetic rats, resulting in the down-regulation of MMP-9 gene expression and enhancing wound healing ability [189]. Poly lactic-glycolic acid NPs (PLGA NPs) encapsulating siRNA for administration to the mucosa of the mouse reproductive tract can effectively improve the survival rate after HSV-2 infection [190]. Additionally, novel hybrid lipopolymer NPs (hNPs) composed of PLGA and dipalmitoyl phosphatidylcholine (DPPC) can deliver siRNA to the lungs and have great potential for the treatment of severe lung diseases [191].

In addition, the efficacy of delivery is related to the strong affinity of endosomes and the weak affinity of cytoplasm for siRNA [192]. Encapsulating micellar NPs (MNPs) with palmitic-acid-linked siRNA (siRNA-PA) can enhance the efficacy of siRNA [193]. Among six complexes, mikto star polymer containing a hydrophobic poly (butyl methacrylate) block could deliver siRNA more effectively to better induce gene silencing [194]. Furthermore, cationic poly(caprolactone)-based (PCL) NPs could be synthesized and stably combined with anionic siRNA and delivered effectively [195]. Combination with palmitic acid could improve the loading efficiency of NPs and the intracellular uptake of siRNAs and at the same time can increase their intracellular half-life [193]. The poly (butyl methacrylate) block can ensure a better silencing ability of siRNA in serum-free medium, while poly(caprolactone) can maintain the stability of siRNAs under physiological conditions and load and deliver intact siRNAs to the cytoplasm [194,195]. Therefore, the addition of some cofactors may improve the delivery efficiency and gene silencing effect of siRNA PNPs.

To evaluate the effect of siRNA PNPs against SARS-CoV-2 in vitro and in vivo, Khaitov et al. identified the most effective siRNA (siR-7) from 15 candidate siRNAs. After adding locked nucleic acids (LNAs) to enhance its stability, it was assembled with peptide dendrimer KK-46 (to form siR-7-EM/KK-46 complex). This siR-7-EM/KK-46 complex was delivered by inhalation, which reduced viral titers and lung inflammation in vitro and in vivo [118]. Therefore, designing polymer nanocarriers with ACE2 antibodies to wrap siRNAs with therapeutic potential can accurately target cells with ACE-2 and exert gene-silencing effects [196].

### 3.3. Glycogen NPs (GNPs) in siRNA Therapeutics

GNPs are randomly branched dendritic nanopolysaccharides composed of glucose repeating units connected by linear α-d -(1-4) glycosidic bonds and α-d -(1-6) branched chains. The glycogen used to prepare GNPs comes from a wide range of sources, from animal tissues to plants (including rabbit and bovine liver, oysters, and sweet corn) [197].

As a functional biomaterial, glycogen has several obvious advantages, including biocompatibility, biodegradability, and high water solubility. These characteristics highlight the potential of GNPs as drug delivery carriers. The outer surface of GNPs is highly branched and has a dense inner core. Because of their high hydrophilicity and biocompatibility, they can easily be biochemically modified to produce functional derivatives [198], such as adhesive GNPs composed of lipoate-conjugated phytoglycogen (L-PG) [199] and oyster GNPs with poly(N-isopropylacrylamide) (PNIPAM) chains on their surface [200].

GNPs are composed of single-molecule spheres (β-particles) and many smaller particles (α-particles). The diameter of GNPs is generally about 20–150 nm and can be modified into various sizes [201]. Cavalieri et al. have found that galacto-GNPs showed a high affinity for prostate cancer cell membranes and could induce the aggregation of prostate cancer cells [202]. They also constructed soft GNPs with a diameter of 20 nm which can deliver siRNAs in human prostate cancer cells more effectively and lead to a 60% gene silencing effect [203]. The size of glycogen is important for the stability of NPs loaded with siRNA as well, as GNPs with a smaller size and a higher protein content are more likely to maintain stability [204].

Gold GNPs were found to deliver siRNA to the lungs in vivo, which can induce a tumor size reduction of nearly 80% in B6 albino mouse lungs grafted with CMT 167 adenocarcinoma cells [205]. Another study applied lipopolysaccharide-amine nanopolymersomes loaded with specific siRNA (siNoggin), which enhanced the effect of bone differentiation in vitro [206]. The construction of cationic enzymatically synthesized glycogen (cESG) can achieve the simultaneous delivery of tetraphenylporphine sulfonate (TPPS) and siRNA, which can reduce the survival rate of cancer cells (ovarian clear cell cancer cells) [207]. In addition, cells transfected with the polysaccharide derivative DMAPA-Glyp/siRNA inhibited the activation of nuclear transcription factor-κB (NF-κB) in human retinal pigment epithelial (hRPE) cells, resulting in the expression of mRNA and protein being reduced, which can be used for gene therapy for diabetic retinopathy [208]. In addition, GNPs have the proven potential to be used in drug-assisted tumor treatment [209]. Therefore, the value of GNPs in COVID-19 needs further exploration.

### 3.4. Other NP Platforms in siRNA Therapeutics

As well as organic NPs, there are also some inorganic NP platforms, mainly including gold NPs (Au NPs) and magnetic NPs (Mag NPs) [123,210]. Au NPs are widely used in diagnosis and treatment and have the potential to be used as drugs and vaccine carriers [211]. They can be used for intranasal delivery and can spread to lymph nodes to activate CD8+ cell-mediated immune responses [211]. It is reported that Au NPs carrying hydrophilic-b-cationic copolymers (HbC-Au NP) could effectively deliver siRNA to cancer cells [210]. siRNAs are located on the surface of Au NPs by gold–thiol chemistry or electrostatic interactions [212]. Au NPs could also easily trigger the immune system through the internalization of APC, thereby presenting outstanding potential in vaccine development [123]. Another study used antigen-coated Au NPs through the principle of plasmon resonance to assist in the detection of the N protein of SARS-CoV-2 [213]. Au NPs could be also used as nanoprobes to detect the S protein of SARS-CoV-2 [214].

Further, Mag NPs with surface hyaluronan can selectively bind to the cell surface adhesion molecule CD44, which plays an important role in the occurrence and development of atherosclerotic plaques, thereby assisting in the magnetic resonance imaging of plaques [215]. Mag NPs can also be used to assist in the detection of the S protein [216]. In addition, layered double hydroxide (LDH) is considered a biocompatible inorganic NP. Studies have found that LDH carrying shRNA can protect it from degradation and make it more easily internalized by cells [217,218]. It has been also found that extracellular vesicles (EVs) have the potential to transport and load antiviral agents into cells to assist in the treatment of COVID-19 [219]. Besides, although a few newly synthesized NPs, such as zinc oxide NPs (ZnO NPs) and titanium dioxide NPs (TiO_2_ NPs), are weakly cytotoxic to host cells, they have been recently applied as a powerful disinfectant to combat SARS-CoV-2 due to their potent antiviral activity [220,221].

Despite all of the advances in vaccines and drugs, the global epidemic of COVID-19 continues. The mutation of the target sequences of RNAi could lead to viral escape [222]. Therefore, designing specific small RNAs for highly conserved viral genome regions could resist viral infections with high mutation rates, which is considered one of the advantages of RNAi therapy [223]. Additionally, multiplexing or combining siRNA targets in a siRNA combination therapeutic could increase protection by providing redundancy, which means that if one target does mutate, the other target will still provide protection. This combination treatment approach has been utilized for HIV-1 to address the highly diverse virus sequences and subtypes globally [224]. Given the rapidly evolving SARS-CoV-2 genome and resulting resistance to vaccines and some monoclonal antibodies, further research is needed to explore and optimize RNAi treatment strategies and appropriate NP carrier platforms against SARS-CoV-2 infection.

## 4. Challenges of Local Delivery of NP-siRNAs—The Importance of Administration Route

The intravenous route can deliver therapeutic siRNAs directly to the liver via the circulatory system, so therapeutic siRNA plays an important role in the treatment of many systemic diseases. Indeed, previous studies on NP carriers have focused on liver delivery [225]. Currently, all four siRNA drugs approved by the FDA are effectively delivered to hepatocytes by GalNAc coupling or LNP carriers for therapeutics [57,62,63,64]. In contrast, local delivery of therapeutic siRNAs to specific organs, such as the lungs, can be achieved via other routes of administration, such as inhalation. Organ-targeted therapy for diseases that may cause damage to specific organs or tissues other than the liver, such as the lungs for COVID-19, is both a potential advantage as well as a challenge for local delivery [226]. Local administration has many advantages, including the rapid onset of action, reduced dosage, and limited side effects [226]. Additionally, since siRNAs used systemically can accumulate in off-target tissues [227], local drug delivery strategies could avoid the off-target accumulation of siRNAs [228].

The primary consideration in the deposition of particles after entering the respiratory tract through the mouth or nose mainly depends on their size and density, with deposition in the lungs occurring through three different ways: impact, sedimentation, and diffusion [229]. As a general rule, aerosols with an aerodynamic diameter >10 µm will deposit in the nasal cavity and pharynx through impact and cannot enter the lower respiratory tract. Particles between ~1 and 10 µm will be deposited in the bronchi and bronchioles through sedimentation, while particles with diameters ranging between 10 and 500 nm will deposit in the lower bronchioles and alveoli by diffusion [230] (Figure 4).

The NP carrier platform could provide many benefits for lung mucosal drug delivery [233]. Several researchers have developed NP-based nasal delivery, aiming to effectively and safely deliver small nucleic acid fragments with therapeutic or vaccine effects to target cells [234]. Most animal experiments focus on intranasal instillation [235] and orotracheal administration [236] to detect the efficacy of NP-siRNAs [236,237,238] (Table 3).

Since SARS-CoV-2 infection is acquired through breathing and mainly occurs in the epithelial cells of the respiratory tract, lipids, polymers, or lipid–polymer hybrids can be configured as inhalable aerosols for the delivery of therapeutic siRNAs [196]. As traditional LNPs tend to accumulate in the liver after IV administration, changing the route of administration from IV to inhalation could prevent NP-siRNAs from accumulating in the liver and instead specifically localize them in the lungs (Figure 4) [248]. An effective measure is to combine nanotechnology with inhalation delivery, where NP-siRNAs are made into dry powders and then delivered to the lung through the nebulization of NP suspensions for local administration. Indeed, lipo-polymeric NPs (LP NPs) can deliver siRNA targeting the ABCC3 gene to the lungs via dry powder inhalation [249]. Freeze-dried pulmonary surfactant (Curosurf^®^)-coated nanogels could also deliver therapeutic siRNAs to the lungs via a nebulizer [250,251]. Another nanocarrier made of triphenyl phosphonium (TPP)-modified generation 4 poly(amidoamine) (PAMAM) dendrimer (G4NH2-TPP) can be made into inhalation preparations after loading siRNA [252]. Thus, local administration of siRNA therapeutics to the lungs can effectively combat lung infections, lung cancer, cystic fibrosis, asthma, and many other diseases of the respiratory system [253].

The most prominent feature of the respiratory tract is that it has several branches. From the outside to the inside are: the nasal cavity, trachea, bronchi, bronchioles, and alveoli. Studies have shown that most lung cancer lesions appear in the bronchi and bronchioles [254], while SARS-CoV-2 infection mainly occurs in pneumocytes and ciliated cells and can cause alveolar damage [255]. The main challenges of pulmonary delivery include mucociliary clearance, alveolar macrophages, the mucus barrier, and shear forces in the nebulizer [256]. The cilia of the lung epithelium can remove particles deposited on it and cause them to be coughed up from the respiratory tract [257]. Additionally, macrophages in the alveoli endocytose and degrade particles through phagocytosis [258]. The presence of mucus in the respiratory system is a factor that seriously affects the transportation of nanoparticles. Mucus covering the respiratory tract, composed of mucin, acts as a physical barrier, which reduces the penetration and diffusion of drugs when it increases in viscosity, preventing the exposure of the respiratory epithelium [257,259]. In addition, pulmonary surfactant (PS) is another barrier that nanoparticles need to pass through before they reach lung cells [260]. Moreover, shear forces increase nebulization temperature and degrade siRNAs [256]. New treatment strategies based on NP-siRNAs provide new possibilities for the treatment of lung diseases and promote research progress on targeted drug delivery to the lung and respiratory tract [261]. Therefore, it is necessary to optimize the size of NPs and apply appropriate excipients to ensure that NP-siRNAs enter target cells more effectively.

In addition to the liver and lung, other potential targets of NP-siRNAs include the eyes, brain, kidney, spleen, and tumor tissues. In a rat spinal cord injury model, injection of NP-encapsulated therapeutic siRNA (PLK4-siRNA) into the injured spinal cord improved motor function to a certain extent [262]. In rats with optic nerve transection, the eye nerves are protected by the delivery of caspase-2-siRNA by intravitreal administration [263]. Therefore, the current delivery strategies vary for different organs.

## 5. Methods to Perform Respiratory Delivery of NP-siRNAs

There are three potential methods to deliver NPs directly to the respiratory tract: intratracheal, intranasal, and orotracheal. Accordingly, inhalation devices are divided into different types, namely nebulizers, pressurized metered dose inhalers (pMDIs), dry powder inhalers (DPIs), and nasal sprays. (Figure 5). Traditional pulmonary drug delivery devices, such as capillary aerosol generators, and recent inhalers, such as Respimat, were mainly applied to deliver micron-sized particles [264,265]. The onTarget system, which is identified as a targeted inhalation device, can also be used to target pharmaceutical ingredients to specific parts of the respiratory tract [266]. An endotracheal aerosolization device (HRH MAG-4) applied a solid LNP suspension as the drug carrier. After verification, it was found that the in vitro efficiency was as high as 90%, and the in vivo efficiency in rats was as high as 80% [267].

Inhalation is the most popular method of targeted lung delivery [260]. There are distinct advantages and disadvantages to using specific inhaler devices to deliver siRNA. Nebulizers are easy to use, requiring only tidal breathing, and enable the delivery of large doses, but they are relatively time-consuming (~5 min) [268] compared to other inhalation devices; furthermore, the presence of shear forces due to their mechanism of action could make them impracticable for the delivery of siRNA since shear stress can degrade nucleic acids, which would have an impact on their biological activity [256]. The advantage of pMDIs is their flow independence; however, their disadvantages are linked to the breath coordination required by the patient for successful delivery. DPIs are the most recently developed delivery device. Comparatively, they are easier to use than the pMDIs since they are breath-actuated. They also contain no propellants and are small and portable, with the only disadvantage being the higher required respiratory rate that makes DPIs not ‘user-friendly’ for pediatric and elderly populations [268]. Regardless of the delivery devices used, nanocarriers are needed to protect siRNA from physical and chemical degradation.

## 6. Methods to Assess Lung Delivery

Inertial impaction methods are used as the gold standard for in vitro assessment of the aerodynamic deposition of inhaled formulations. Data from these methods are regulatory requirements to estimate the amount of the drug that is delivered to the lungs [269]. These methods classify particles according to the aerodynamic diameter of the entire delivered dose [270]. An impactor consists of a series of nozzles (circular or slot-shaped) and an impaction surface [271,272]. Cascade impactors operate on the principle of curvilinear motion of particles in the aerosol stream. Air is drawn into the impactor using a vacuum pump, and the air stream flows through the nozzles and toward the impaction surface, where particles are separated from the air stream by their inertia. Larger particles collect on the impaction surface, while small particles do not and follow the air stream. Several types of impactors are used to assess delivery to the lungs, such as the Andersen cascade impactor (ACI, Figure 6A) used by the United States Pharmacopeia (USP) (Table 4).

A pre-separator is typically used between the induction port and stage 0 of a cascade impactor to capture the large, non-inhalable carrier particles in DPI formulations to prevent impactor over-loading. By analyzing the amount of active drug deposited on various stages in which the geometrical models are individually built up for the whole dimensions of the ACI, it is then possible to calculate several essential parameters that define the overall inhalation formulation performances. These are the fine particle dose (FPD) and fraction (FPF), mass median aerodynamic diameter (MMAD), and geometric standard deviation (GSD) of the active drug particles collected. The impactors are used to assess the performance of pMDIs, nebulizers, and DPI devices. Using a calibrated pump, these devices operate at specific flow rates for each that simulate a patient lung capacity of 4 L; specifically, 28.3 L/min for pMDIs or at 60 and 100 L/min with modifications for DPIs [273].

Another type of cascade impactor is the next-generation pharmaceutical impactor (NGI, Figure 6D), which was specifically designed for pharmaceutical testing. Particles are deposited on seven stages (collection cups) that are held in a removable tray for sample collection. It also has a micro-orifice collector (MOC) that captures extremely small particles in a collection cup, normally collected on the final filter in other impactors [274]. NGIs are ideal for characterizing drug deposition by nebulizers [275].

For the assessment of DPIs, the multi-stage liquid impinger (MSLI, Figure 6B) can also be used. The MSLI has four impaction stages, each containing a small volume of liquid during operation, and a final filter stage. The liquid in the impinger reduces particle bounce. For the operation of the MSLI, the induction port is connected to the inhaler, but unlike the ACI’s, they do not require a pre-separator.

The twin stage impinger, a two-stage separation device (Figure 6C), is a more ‘elementary’ impinger that separates particles according to two different aerodynamic cut-off sizes only. The aerosol is fractionated by rushing through a simulated oropharynx and then through an impinger stage of defined aerodynamic particle size cut-off characteristics. The fine (pulmonary) fraction which penetrates is collected by a lower impinger. The twin impinger is a valuable device for routine quality assessment of aerosols during product development, stability testing and quality assurance, and the comparison of inhalation drugs [276].

Another two-stage impactor is the fast-screening impactor (FSI, Figure 6E), which incorporates the NGI’s pre-separator technology and utilizes the same induction port. The FSI first separates large non-inhalable particles which are captured by the liquid trap, followed by an impaction stage at a 5-micron cut-off (30–100 L/min). The fine particle dose is collected on a glass fiber filter. Additional inserts are available for 10-micron cut-off size, but only at a 30 L/min flow rate. When used for DPI deposition assessments, FPF performance is comparative to that of full NGI, and the FPD is comparative or better when using the FSI, depending on the drug tested [277].

While these impactors are an excellent way to test aerodynamic particle size distribution in the lungs, other aspects of the drug uptake, such as dissolution rate, transport through the epithelium, and therapeutic efficacy of the drugs, cannot be determined using standard impaction methods [278,279]. To address this issue, novel hybrid approaches and modifications have been developed, which involve marrying the biological representation of the lung epithelium with the impaction instrumentation.

Air–liquid interface (ALI) culture models are an effective way to produce epithelial layers that represent the pulmonary epithelium [280]. Specifically, the Calu-3 lung cell line has been a promising in vitro model of airway epithelia due to its similarity to in vivo physiology [281]. Nevertheless, the use of primary broncho-epithelial cells is the gold standard and can be obtained from donors who are healthy or have diseases and/or may be infected with viruses [280,282,283,284]. Using these epithelial layers, assays that test the biochemical characteristics of drugs, such as permeability, integrity, and mucus production of epithelial layers, can all be performed. Additionally, the impact of the deposited drugs on inflammation and wound healing can be investigated [285,286]. Haghi et al. (2014b) developed a modified Andersen cascade impactor (mACI) that incorporated ALI models of Calu-3 cells inserted within the mACI at stages 4–7, representing the deep lung region. These modified inserts can be customized to be inserted at various stages of the ACI for studies in both the upper and the lower respiratory regions of the lungs. By integrating the ALI culture models within the ACI, the deposition and subsequent permeability of formulations delivered with pMDIs and DPIs can be assessed with physiological relevance [279,287,288].

Similarly, the NGI can also be modified to include an ALI cell culture model (mNGI), enabling aerosol deposition and transport across cell epithelia. mNGIs have been engineered to deliver aerosolized particles under controlled vacuum flow to ALI-cultured Calu-3 cells at stages 2, 3, and 4 [289] and stages 3, 5, and 7 [290], highlighting the versatility and adaptability of an mNGI depending on the research question.

Thus, utilizing standard and modified cascade impactor methods enables a thorough assessment of drug deposition. Both the aerodynamic distribution in the lungs as well as cellular responses such as uptake, transport, and therapeutic efficacy of known and novel inhaled formulations can be determined.

## 7. Animal Models to Assess Pulmonary Delivery

Preclinical animal models are essential for the translation of RNAi-based therapeutics from in vitro studies to human clinical implementation [291]. Several factors need to be considered when choosing appropriate animal models. Relative costs and logistics with the handling and storage of animals are major factors dictating the species used in preclinical studies.

Mouse models are the most widely used animal models to assess the pharmacokinetics and the effects of pulmonary drug delivery [291]. They reflect important features of human disease with severe infection and virus-induced hyper-inflammation, as well as features of acute lung injury, acute respiratory distress syndrome, impaired lung function. Mice are relatively inexpensive and easy to handle and can be bred in large quantities for high-throughput analyses [40]. Larger rodent species, such as rats or hamsters, are also used for the examination of pulmonary drug delivery; however, these models are much less commonly used compared to standard mouse models. This is due to their larger size and lack of established infrastructure in many laboratories, which dictates higher general costs for purchasing, larger specialized housing requirements, and ultimately, smaller sample size. Similarly, ferrets are less commonly used than mouse models due to higher costs and the requirement for unique housing arrangements. This species shares a high degree of anatomical similarity with the human respiratory tract, making them an excellent model for pulmonary drug delivery assessment [292]. Non-human primates (NHP), such as the Rhesus macaque, are uncommonly used for pulmonary drug delivery assessment and are typically only used in the final stages of toxicology and safety profiling prior to human trials [293]. NHP models are very expensive and require highly specialized animal facilities, which are rare. NHP models have a very high fidelity of recapitulation of the human respiratory system, making them the ‘gold standard’ in terms of clinical translatability. Within the context of SARS-CoV-2 infection, a summary of the different animal models to interrogate COVID-19 pathogenesis can be found here [291].

Routes of pulmonary drug delivery are relatively similar between all different animal models; however, the volume of the drug administered and the specialized equipment required all need to be scaled to the relative size of the animal species [278]. Intranasal drug delivery is the most common pulmonary drug delivery system and can be performed using light anesthesia to administer small volumes to the nostrils, which can be done quickly and with relative ease. However, particle size is an important determining factor for successful intranasal drug delivery, as it needs to bypass the mucociliary barrier within the nasopharynx for deposition into the distal airways and alveoli [294]. Comparatively, intratracheal drug delivery requires the induction of surgical anesthesia prior to tracheostomy and/or tracheal intubation to administer larger volumes to the lungs, which is more labor-intensive and takes longer compared to intranasal delivery [295]. Aerosol exposure is often used to deliver nebulized drug formulations that are suspended in a particulate state. This is often achieved using a passive inhalation system via nose-only exposure or a whole-body exposure chamber. This is less common, largely due to low throughput capacity, variable dosage delivery, larger amounts of the drug being required, and the process is stressful for the animal due to the restraints used during the procedure [278,296].

Administration of RNAi-based therapeutics in mice via the intranasal and intratracheal routes are highly successful for the amelioration of experimental asthma disease phenotypes [297,298,299,300,301,302], chronic obstructive pulmonary disease, lung cancer, influenza, and SARS-CoV infections [298,303,304,305]. Recently, lipid-nanoparticle-encapsulated siRNA formulations injected via intravenous administration were shown to increase survival and significantly reduce infectious viral titers in a lethal model of SARS-CoV-2 infection in K18-hACE2 mice [30]; however, the administration of these formulations via pulmonary delivery was not evaluated. These mice have the human ACE2 receptor incorporated into the epithelial specific K18 promoter and thus express it on the epithelial surface to enable efficient infection [291]. Furthermore, intranasal delivery of siRNA to block SARS-CoV-2 RdRp significantly enhanced the antiviral response and reduced lung inflammation in a COVID-19 hamster model [118]. The implementation of siRNA therapies for pulmonary delivery in both rats and ferrets has been examined in the context of asthma, chronic obstructive pulmonary disease, cystic fibrosis, and influenza infection models; however, to date, this has not been explored for COVID-19 [306,307,308]. Finally, the use of siRNA in NHP was highly effective in post-exposure protection against the Ebola virus when given intravenously, whilst prophylactic and therapeutic siRNA intranasal administration reduced SARS pathogenesis in Rhesus macaques, providing strong rationale and incentives for the use of NHP in developing RNAi-based therapeutics for COVID-19 [309,310].

## 8. Conclusions

Despite the advances in RNA therapeutics and nanomedicine, COVID-19 continues to cause severe infections and deaths globally, highlighting the urgent need to optimize RNAi treatment strategies and appropriate NP carrier platforms for SARS-CoV-2 infection. Importantly, these future advances will provide preparedness in the form of lung therapeutic delivery strategies for future emerging diseases.

## Figures and Tables

**Figure 1 ijms-23-02408-f001:**
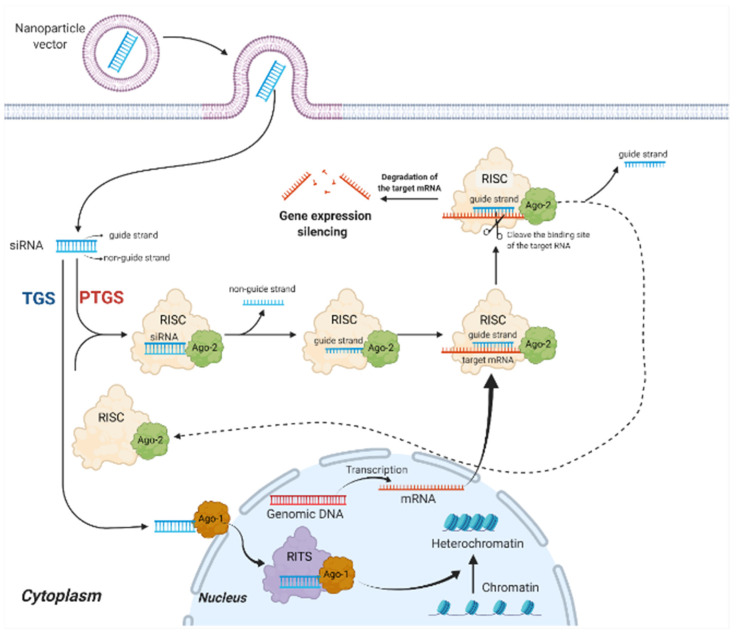
Mechanisms of siRNA therapy. PTGS primarily occurs in the cytoplasm. The guide strand binds to Ago-2 of RISC, while the non-guide strand is cleaved and ejected by the Ago-2; Ago-2 carries the guide strand to target mRNA, causing gene expression silencing. TGS has taken place in the nucleus. siRNA binds to Ago-1 on RITS, which contributes to the production of heterochromatin and induces the epigenetic silencing of the gene. PTGS, post-transcriptional gene silencing pathway; TGS, transcriptional gene silencing; Ago-1, Argonaute-1 protein; Ago-2, Argonaute-2 protein; RISC, RNA-induced silencing complex; RITS, RNA-induced transcriptional silencing complex. Created with BioRender.com.

**Figure 2 ijms-23-02408-f002:**
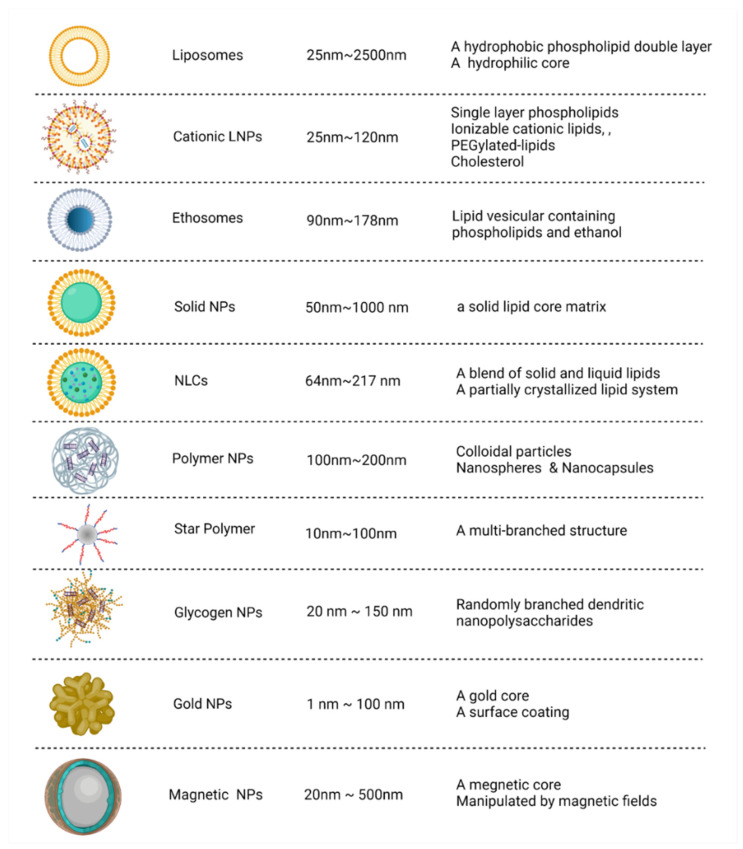
Comparison of different nanocarriers. Advanced nanocarriers can be divided into organic NPs (lipid NPs, polymers NPs, and glycogen NPs) and inorganic NPs (gold NPs and magnetic NPs) according to their size, structures, and characteristics. NPs, nanoparticles; NLCs, nanostructured lipid carriers; LNPs, lipid nanoparticles. Created with BioRender.com.

**Figure 3 ijms-23-02408-f003:**
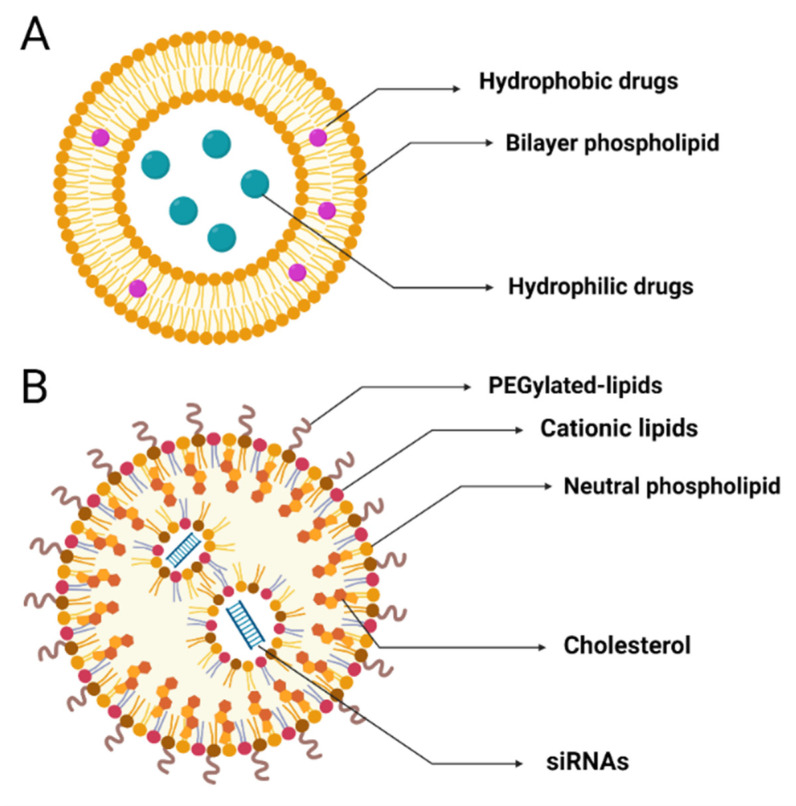
Structures of liposomes and cationic LNPs. (**A**) Liposomes: small-molecule artificial vesicles with a hydrophobic phospholipid double layer outside and a hydrophilic core inside. (**B**) Cationic LNPs: spherical particles composed of ionizable cationic lipids, cholesterol, phospholipids, and PEGylated lipids. LNPs, lipid nanoparticles; PEG, polyethylene glycol. Created with BioRender.com.

**Figure 4 ijms-23-02408-f004:**
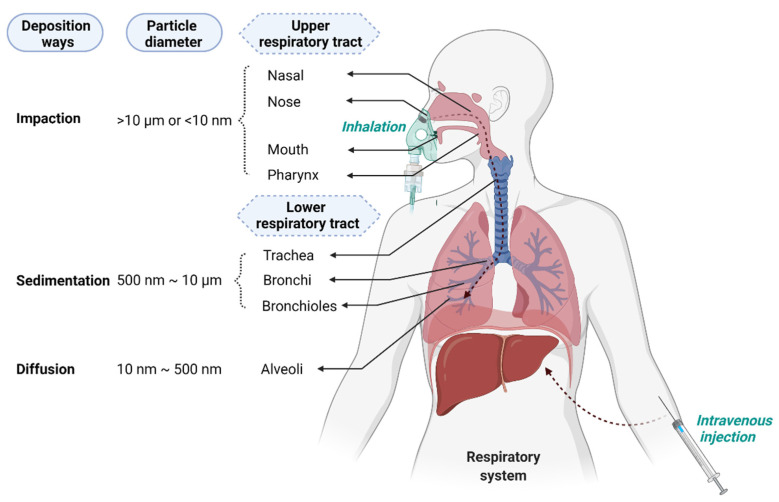
Intravenous delivery vs. lung delivery of NP-siRNAs. Intravenous delivery can deliver therapeutic siRNAs directly to the liver via the circulatory system. Lung delivery can deliver siRNAs into the respiratory tract through the mouth or nose, and the deposition distance depends mainly on the size and density of the particles. Aerosols with an aerodynamic diameter >10 µm will deposit in the nasal cavity and pharynx through impact, and particles with a diameter of ˂10 nm with high diffusion velocity are more likely to be deposited in the upper respiratory tract [231,232]. Particles between ~1 and 10 µm will be deposited in the bronchi and bronchioles through sedimentation; particles with diameters ranging between 10 and 500 nm will deposit in the lower bronchioles and alveoli by diffusion. Created with BioRender.com.

**Figure 5 ijms-23-02408-f005:**
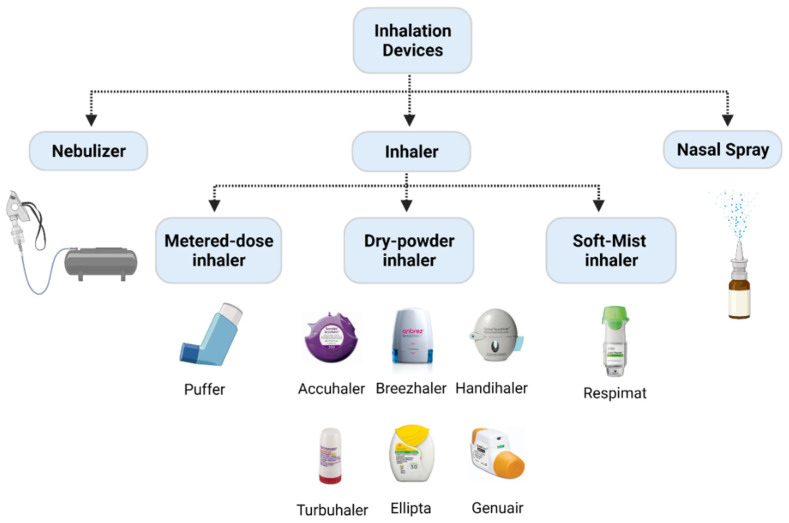
Examples of different inhalation devices. Inhalation devices are divided into nebulizer, inhaler (including pMDIs, DPIs, and SMIs), and nasal spray. pMDIs, pressurized metered dose inhalers; DPIs, dry powder inhalers; SMIs, soft mist inhalers. Created with BioRender.com.

**Figure 6 ijms-23-02408-f006:**
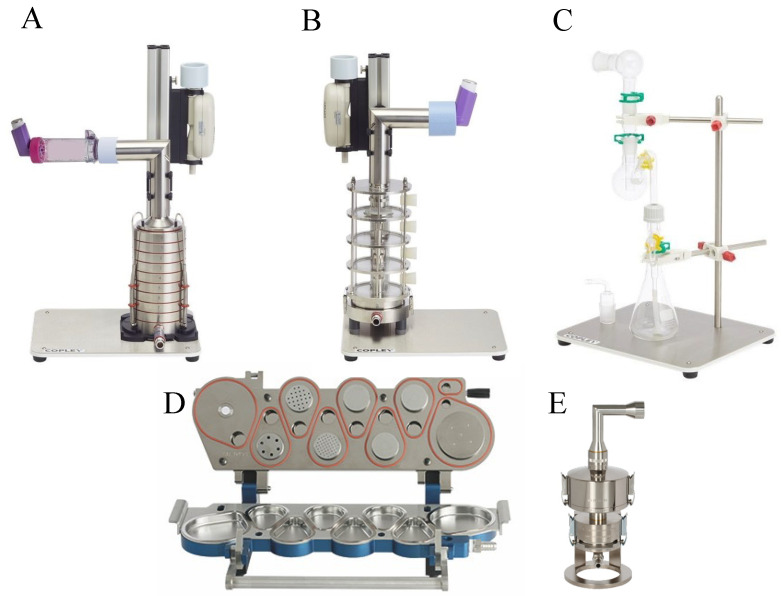
Types of impactors. (**A**) Andersen cascade impactor; (**B**) multi-stage liquid impinger; (**C**) twin-stage impinger; (**D**) next-generation impactor; (**E**) fast-screening impactor. Adapted from images from www.copleyscientific.com.

**Table 1 ijms-23-02408-t001:** siRNA therapeutics for SARS-CoV-2, SARS-CoV-1, and MERS-CoV.

Virus	Sequence (5′-3′)	Region	Stage	Refs
MERS-CoV	UAGAAGAACAGCUAUCACCCU	ORF1ab	Computational approach	[99]
MERS-CoV	AACAUAGAAAGCAGAUAGGUC	ORF1ab	Computational approach	
MERS-CoV	UCUAAGAGCUGCAUAAGUGUC	ORF1ab	Computational approach	
MERS-CoV	AUCAAGAAAAGCGUUAGAGGA	ORF1ab	Computational approach	
MERS-CoV	UUUGUAGUACCAAUGACGCAA	ORF1ab	Computational approach	
MERS-CoV	UAAUAGUAAAAAUAGAUUGCU	ORF1ab	In vitro (HEK-293)	[108]
MERS-CoV	AACAUUAAUAGCAUUAUCCAU	ORF1ab	In vitro (HEK-293)	
MERS-CoV	UAAGAUAUCAUCUAAAGUGUC	ORF1ab	In vitro (HEK-293)	
MERS-CoV	UUAAAACUCAAACUAAUAGCA	ORF1ab	In vitro (HEK-293)	
MERS-CoV	UAGUUAAAGAGUUUCUAAGAG	ORF1ab	In vitro (HEK-293)	
MERS-CoV	UAAUAGUAAAAAUAGAUUGCU	ORF1ab	In vitro (Vero cells)	[109]
MERS-CoV	UAAGAUAUCAUCUAAAGUGUC	ORF1ab	In vitro (Vero cells)	
MERS-CoV	AACAUUAAUAGCAUUAUCCAU	ORF1ab	In vitro (Huh7)	[110]
MERS-CoV	UUAAAACUCAAACUAAUAGCA	ORF1ab	In vitro (Huh7)	
MERS-CoV	AUUAAAUCUGUUAAUGUUGUU	ORF1ab	In vitro (Huh7)	
MERS-CoV	AAAUAGUUAUGAAUAGUUGAG	ORF1ab	In vitro (Huh7)	
SARS-CoV-1	GGGCUAUCAACCUAUAGAU	Spike protein (S)	In vitro (Vero E6)	[111]
SARS-CoV-1	CAAGGCGAUUAGUCAAAUU	Spike protein (S)	In vitro (Vero E6)	
SARS-CoV-1	CGUAACUAAACAGCACAAG	3′-UTR	In vitro (Vero E6)	
SARS-CoV-1	GCUCCUAAUUACACUCAAC	ORF2, spike	In vitro (FRhk-4)	[112]
SARS-CoV-1	GGAUGAGGAAGGCAAUUUA	ORF1b, nsp-12	In vitro (FRhk-4)	
SARS-CoV-1	GGAUAAGUCAGCUCAAUGC	ORF1b, nsp-13	In vitro (FRhk-4)	
SARS-CoV-1	CUGGCACACUACUUGUCGA	ORF1b, nsp-16	In vitro (FRhk-4)	
SARS-CoV-1	GCUCCUAAUUACACUCAAC	ORF2, spike	In vivo (Rhesus macaque)	[113]
SARS-CoV-1	GGAUGAGG AAGGCAAUUUA	ORF1b, nsp-12	In vivo (Rhesus macaque)	
SARS-CoV-1	GUACCCUCUUGAUUGCAUC	Replicase 1A	In vitro (FRhk-4)	[114]
SARS-CoV-1	GAGUCGAAGAGAGGUGUCU	Replicase 1A	In vitro (FRhk-4)	
SARS-CoV-1	GCACUUGUCUACCUUGAUG	Replicase 1A	In vitro (FRhk-4)	
SARS-CoV-1	CACUGAUUCCGUUCGAGAUC	S	In vitro (FRhk-4)	[115]
SARS-CoV-1	CGUUUCGGAAGAAACAGGUAC	E	In vitro (FRhk-4)	
SARS-CoV-1	UGCUUGCUGCUGUCUACAG	M	In vitro (FRhk-4)	
SARS-CoV-1	GUGGCUUAGCUACUUCGUUG	M	In vitro (FRhk-4)	
SARS-CoV-1	UGAAGGAGUUCCUGAUCUUCU	Small envelope protein (E)	In vitro (Vero E6)	[116]
SARS-CoV-1	AGCUUAAACAACUCCUGGAAC	Small envelope protein (E)	In vitro (Vero E6)	
SARS-CoV-1	GAUAAUGGACCCCAAUCAAAC	Membrane protein (M)	In vitro (Vero E6)	
SARS-CoV-1	CUUACAUAGCUCGCGUCUC	Nt 14450 to 14468	In vitro (Vero cells)	[117]
SARS-CoV-1	GAAUAUUAGGCGCAGGCUG	Nt 15877 to 15895	In vitro (Vero cells)	
SARS-CoV-2	UUCGUUUAGAGAACAGAUC	5′-UTR	In vitro (Vero E6)	[34]
SARS-CoV-2	GUACUUUUUUUGAACUUCUACA	Surface glycoprotein	In vitro (Vero cells)	[91]
SARS-CoV-2	CAACAAAGAUAGCACUUAA	ORF1ab polyprotein	In vitro (Vero cells)	
SARS-CoV-2	UCAUACCACUUAUGUACAA	ORF1ab polyprotein	In vitro (Vero cells)	
SARS-CoV-2	CCAAAAUCAUAACCCUCAAA	ORF3a protein	In vitro (Vero cells)	
SARS-CoV-2	AAACCUUCUUUUUACGUUUA	Envelope protein	In vitro (Vero cells)	
SARS-CoV-2	CGAACGCUUUCUUAUUACAA	Membrane glycoprotein	In vitro (Vero cells)	
SARS-CoV-2	AUGAACCACAAAUCAUUACUA	S	In vitro (human HUVECs and A549 cells)	[104]
SARS-CoV-2	AGAUCAACUUACUCCUACUUG	S	In vitro (human HUVECs and A549 cells)	
SARS-CoV-2	AUAUAAUUCCGCAUCAUUUUC	S	In vitro (human HUVECs and A549 cells)	
SARS-CoV-2	AAGUCAAACAAAUUUACAAAA	S	In vitro (human HUVECs and A549 cells)	
SARS-CoV-2	AAUACACCAAAAGAUCACAUU	N	In vitro (human HUVECs and A549 cells)	
SARS-CoV-2	UAUUGACGCAUACAAAACAUU	N	In vitro (human HUVECs and A549 cells)	
SARS-CoV-2	AAGGAACUGAUUACAAACAUU	N	In vitro (human HUVECs and A549 cells)	
SARS-CoV-2	GUUCCAAUUAACACCAAUAGC	N	In vitro (human HUVECs and A549 cells)	
SARS-CoV-2	AACUAUAAAUUAAACACAGAC	M	In vitro (human HUVECs and A549 cells)	
SARS-CoV-2	AAACUAACAUUCUUCUCAACG	M	In vitro (human HUVECs and A549 cells)	
SARS-CoV-2	CUAUUCCUUACAUGGAUUUGU	M	In vitro (human HUVECs and A549 cells)	
SARS-CoV-2	GUUGGACUGAGACUGACCUUA	RdRp	In vitro (Vero E6) and in vivo (K18-hACE2 mice)	[30]
SARS-CoV-2	UUAUACCUUCCCAGGUAACAA	5′UTR	In vitro (Vero E6) and in vivo (K18-hACE2 mice)	
SARS-CoV-2	UCACCUUAUAAUUCACAGAAU	Helicase	In vitro (Vero E6) and in vivo (K18-hACE2 mice)	
SARS-CoV-2	GGAAGGAAGUUCUGUUGAA	RdRp	In vitro (Vero cells) and in vivo (Syrian hamsters)	[118]
SARS-CoV-2	UUUGUAUGCGUCAAUAUGCUU	Nucleocapsid phosphoprotein	Computational approach	[119]
SARS-CoV-2	UCAACGUACACUUUGUUUCUG	Surface glycoprotein genes	Computational approach	
SARS-CoV-2	AAAAACUUCACCAAAAGGGCA	Surface glycoprotein genes	Computational approach	
SARS-CoV-2	UUAAAAACUUCACCAAAAGGG	Surface glycoprotein genes	Computational approach	
SARS-CoV-2	UUAAAGCACGGUUUAAUUGUG	Surface glycoprotein genes	Computational approach	
SARS-CoV-2	AACUUCUUGGGUGUUUUUGUC	Surface glycoprotein genes	Computational approach	
SARS-CoV-2	UUUGAUUGUCCAAGUACACAC	Surface glycoprotein genes	Computational approach	
SARS-CoV-2	UAAUUUGACUCCUUUGAGCAC	Surface glycoprotein genes	Computational approach	
SARS-CoV-2	UCCUUCUUUAGAAACUAUACA	ORF1ab	Computational approach	[102]
SARS-CoV-2	UGGUUUCACUACUUUCUGUUU	ORF1ab	Computational approach	
SARS-CoV-2	UUCACUACUUUCUGUUUUGCU	ORF1ab	Computational approach	
SARS-CoV-2	AUGUCAUCCCUACUAUAACUCAAA	ORF1ab	Computational approach	
SARS-CoV-2	UUAAAAUAUAAUGAAAAUGGA	S	Computational approach	
SARS-CoV-2	CUUGAAGCCCCUUUUCUCUAUCUUU	ORF3a	Computational approach	
SARS-CoV-2	UUGAAUACACCAAAAGAUCACAUU	N	Computational approach	
SARS-CoV-2	AGUAGAAAUACCAUCUUGGAC	N	Computational approach	[120]
SARS-CoV-2	GUUUAGAGAACAGAUCUACAA	The leader sequence	Computational approach	[103]
SARS-CoV-2	UAGUACUACAGAUAGAGACAC	RdRp	Molecular dynamics (MD) simulation	[101]

**Table 2 ijms-23-02408-t002:** Lipid nanocarriers loaded with therapeutic siRNAs.

Carrier	siRNAs’ Target	Cells/Animal Models	Refs
Ionizable LNPs	LINC01257 siRNA	Kasumi-1 cells (human acute myeloid leukemia cell line)	[153]
Ionizable LNPs	Irf5 siRNA	C57BL/6 Irf5−/− Apoe−/− mice	[154]
Ionizable LNPs	MRTF-B siRNA	Human conjunctival fibroblasts	[155]
Ionizable LNPs	CTNNB1 Dicer substrate RNA (DsiRNA)	HepG2 and A375.S2 cells	[156]
Ionizable LNPs	RBPMS siRNA	HEK293-GFP stable cells (GFP293)	[157]
Ionizable LNPs (CL4H6)	STAT3 HIF-1α	ICR (♀, 4–10 weeks) mice and BALB/cAjcl-nu/nu (♂, 4–5 weeks) mice	[158]
Ultra-small LNPs	MK siRNA	Human HCC cell line, HepG2, and male BALB/c nude mice	[159]
Ginger-derived NPs (GDNPs)	Dmt1 siRNA	Hamp KO mice	[160]
SLNs	PD-1	J774A.1 murine macrophages and B16-F10 murine melanoma cells	[161]
SLNs	TNF-α	J774A.1 murine macrophages	[162]
SLNs	Bcl-2	HeLa (human cervical cancer adenocarcinoma) cell line	[163]
SLNs	BACE1	Caco-2 cells (human epithelial colorectal adenocarcinoma cells)	[164]
SLNs	MDR1	MCF-7 cells and MCF-7/ADR cells	[165]
SLNs	KSP	Murine endothelial cell line MS1-VEGF and human primary umbilical vein endothelial cell line (HUVEC); human dermal microvascular endothelial cell (HMEC-1) line	[166]
SLNs	Bcl-2	A549 cells (human lung carcinoma)	[167]
SLNs	MCL1	KB cells (human epithelial carcinoma) and BALB/c nu/nu mice	[168]
SLNs	c-Met	U-87MG human GBM cells and U-87MG tumor-bearing mouse	[169]
SLNs	VEGF	PC-3 (human prostate cancer cells) and MDAMB435 (human breast cancer cells)	[170]
Cationic SLNs	EphA2 siRNA	PC-3 and DU145 (prostate cancer cell lines)	[171]
NLCs	BCL2 and MRP1	A549 human lung adenocarcinoma cells	[149]
NLNs	GFP	Chinese hamster ovary cells (CHO-GFP) and human embryonic kidney (HEK293) cells	[150]
Ethosomes	GAPDH	Human adult epidermal keratinocytes and female BALB/c mice	[151]
Echogenic Liposomes	CPP	Human breast adenocarcinoma cells (HT-1080)	[152]

**Table 3 ijms-23-02408-t003:** Mouse models used to assess respiratory nanocarrier delivery.

Mouse Models	Carriers	Drugs	Delivery	Refs
Female BALB/c mice	Solid lipid nanoparticles (SLNs)	Rhynchophylline (Rhy)	Aerosol administration	[239]
Female BALB/c mice	Dextran nanoparticles	SET-M33 (a non-natural antimicrobial peptide)	Hamilton syringe and laryngoscope	[240]
Female BALB/c mice	Dextran nanogels	siRNA	Tracheal aspiration	[241]
Female BLAB/c mice	Nanocluster	Ceftazidime	Dry powder inhalation	[242]
Female CF-1 mice	PEGylated human ferritin nanocages (FTn)	Doxorubicin (DOX)	Inhalation	[243]
Female CF-1 mice	PLGA and PLGA–PEG NP	Dexamethasone sodium phosphate (DP)	Intranasal instillation	[244]
Female C57BL/6OlaHsd mice	PEG–PEI copolymer	Dexamethasone (DEX)	Aerosol administration	[245]
Male CF-1 mice	Mucus-penetrating particles (MPP)	Fluticasone propionate (FP)	Intratracheal instillation	[246]
Male CD-1 mice	Mesoporous silica nanoparticles (MSNs)	Nanopure water droplets	Aerosol nose-only exposure system	[247]

**Table 4 ijms-23-02408-t004:** USP cascade impactors for orally inhaled device testing.

Impactors	Devices	Flow Rates	# of Stages	Particle Size Range
Andersen cascade impactor	pMDIs and DPIs	28.3 L/min (60 and 90 L/min using modifications)	6 or 8	0.4–0.9 microns (28.3 L/min), 0.3–8.6 (60 L/min), 0.2–8 (90 L/min)
Multi-stage liquid impinger	DPIs	30–100 L/min	4	1.7–13 microns
Twin-stage impinger	pMDIs and nebulizers	60 L/min	2	6.4 microns
Fast-screening impactor	pMDIs, DPIs, and nasal spray	30–100 L/min	2	5 and 10 microns
Next-generation pharmaceutical impactor	pMDIs, DPIs, and nebulizers	15–100 L/min	7	0.24–14.1 microns

## Data Availability

Not applicable.

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
