# Peer review of "Nanoparticle Delivery Platforms for RNAi Therapeutics Targeting COVID-19 Disease in the Respiratory Tract"

_ijms, 2022, doi:10.3390/ijms23052408_

Round 1

Reviewer 1 Report

The Review  is  designed very well and professionally and discuss very important issue , which is The use of nanoparticles (NPs) as carriers for the delivery of  small interfering RNA (siRNAs) to specific tissues or organs of the human body that could play a vital role in the specific therapy of severe respiratory infections, such as COVID-19. 

I recommend authors to add part about some nanoparticles as cited in this article ( https://doi.org/10.3390/coatings11040388),(https://doi.org/10.3390/coatings11060680)..these articles will benefit this article concept greatly.

Utilizing of (Zinc Oxide Nano-Spray) for Disinfection against “SARS-CoV-2” and Testing Its Biological Effectiveness on Some Biochemical Parameters during (COVID-19 Pandemic)—”ZnO Nanoparticles Have Antiviral Activity against (SARS-CoV-2)”.
A New Sterilization Strategy Using TiO2 Nanotubes for Production of Free Radicals that Eliminate Viruses and Application of a Treatment Strategy to Combat Infections Caused by Emerging SARS-CoV-2 during the COVID-19 Pandemic.

Author Response

Response: Thanks very much for the recommended articles, which are quite helpful. We have made revisions according to the comment. A related sentence which cites the above two references have been added to lines 476 to 479 as follows:

"Besides, although a few newly synthesized NPs, such as zinc oxide NPs (ZnO-NPs) and titanium dioxide NPs (TiO2-NPs), are weakly cytotoxic to host cells, they have been recently applied as powerful disinfectant to combat SARS-CoV-2 due to their potent antiviral activity [220, 221]."

Revised sentence was marked in blue in the review.

Reviewer 2 Report

The authors give a broad overview about the possibilities of RNA interference-based gene therapies for treatment of Covid-19.  They describe different organ- and cell-specific targeted nanoparticles (NPs) systems delivering small interfering RNA (siRNAs) to specific tissues or organs and cells of the human body. In this review, they first give a summarized presentation of the mechanism of the RNAi therapy, then a number of cell-derived and virus-encoded gene targets, followed by tables containing siRNA sequences for genes of MERS-CoV, SARS-CoV-1 and SARS-CoV-2, and at least variety of novel nanocarriers. They are composed carriers such as Lipid NPs, Star Polymer NPs, and Glycogen NPs, and summarize the pre-clinical/clinical progress of these nanoparticle platforms in siRNA delivery. Many siRNA examples of cellular origins are listed with used lipid containing nano-carriers and those with lipid-free carriers provides supplied by references. The airway application is discussed for human beings and supported by mouse models and operative instrumentations.

The authors provide is a comprehensive review which has turned out well.

A minor recommendation: in lines 212 to 216, different siRNA aims were listed, among them the cellular TMPRSS2. This is correct, but the second important cell protease  furin, which co-operates with TMPRSS2 during virus invasion is missed, and the second protease furin and the corresponding references should be included.

Author Response

Response: Thank you very much for this comment. We have added the second important cell protease furin and the corresponding reference (PMID: 32703818 “TMPRSS2 and furin are both essential for proteolytic activation of SARS-CoV-2 in human airway cells”) to the sentence in lines 215 to 217, as follows:

"Meanwhile, the cell protease furin have been found to activate the S protein of SARS-CoV-2 through cooperation with TMPRSS2 during virus invasion, which makes furin another target of siRNAs [92]."

Revised sentence was marked in Red in the paper.